# Influence Mechanism of Initial Concreting Temperature and Water-Binder Ratio on Hydration Rate of Fly Ash Concrete

**DOI:** 10.3390/ma16124286

**Published:** 2023-06-09

**Authors:** Juanjuan Quan, Shaojun Fu, Jian Chen, Rudong Yang, Tao Luo, Ding Wang

**Affiliations:** 1Shaanxi Key Laboratory of Safety and Durability of Concrete Structure, Xijing University, Xi’an 710123, China; 15862853635@163.com (J.C.); 18851956430@163.com (R.Y.); luotao19870426@126.com (T.L.); 18382342289@163.com (D.W.); 2School of Civil Engineering, Wuhan University, Wuhan 430072, China

**Keywords:** initial concreting temperature, water-binder ratio, adiabatic temperature rise, hydration rate, fly ash concrete

## Abstract

The hydration exothermic rate of fly ash concrete is significantly affected by the initial concreting temperature and water-binder ratio. Firstly, the adiabatic temperature rise and temperature rise rate of fly ash concrete at different initial concreting temperatures and water-binder ratios were obtained by a thermal test instrument; then, the effects of initial concreting temperature and water-binder ratio on the hydration kinetic parameters of the NG-I-D hydration process of fly ash concrete were analyzed by the theory of hydration kinetics; lastly, the effects of initial concreting temperature and water-binder ratio on chemically bound water and pore bulk of fly ash concrete during hydration were analyzed by applying a thermogravimetric analyzer and industrial CT scanning techniques. The results showed that the increase in initial concreting temperature and the decrease in water-binder ratio accelerated the rate of temperature rise, and the initial concreting temperature had a more significant effect than the water-binder ratio. During the hydration reaction, the I process was significantly influenced by the initial concreting temperature, and the D process was significantly influenced by the water-binder ratio; the content of bound water increased with the increase in water-binder ratio and age and the decrease in initial concreting temperature. The initial temperature had a significant effect on the growth rate of 1 to 3 days bound water, and the water-binder ratio had a more significant effect on the growth rate of 3 to 7 days bound water. The porosity was positively correlated with the initial concreting temperature and water-binder ratio and decreased with age, but 1 to 3 days was the key period of porosity change. Additionally, the pore size was also influenced by the initial concreting temperature and water-binder ratio.

## 1. Introduction

Temperature cracking is always one of the most important problems in mass concrete [1,2,3]. The current strategies for the prevention and control of temperature cracks mainly include reducing the hydration heat of cementitious materials (e.g., mixing with fly ash), regulating the concrete temperature field (e.g., cooling) [4], improving the anti-cracking ability of concrete materials (e.g., increasing the tensile strength) [5], and selecting the good structure form for the rational stress distribution (e.g., structural inducing joints, pre-stressing) [3]. The selection of temperature control and crack prevention measures is mainly based on temperature field research on mass concrete structures [1,2,3]. The temperature field of mass concrete is affected by internal and external factors [6,7,8,9,10,11,12,13,14,15,16,17]. Among them, the internal factors of hydration and heat release from cementitious material are the key factors. The hydration heat release process is also affected by concrete material composition, structure, and environmental conditions. Therefore, the degree of understanding of the concrete hydration heat release process is related to the rationality and effectiveness of temperature control measures. It has been proven that the hydration rate of concrete interacts with its own temperature [3], but most of the current adiabatic temperature rise models do not fully consider the influence of the hydration rate, which is semi-empirical and semi-theoretical; even models that take into account the effect of the hydration rate are limited in their application due to poor adaptability and difficulty in determining parameters. Therefore, it is of great significance to study the hydration rate of concrete for temperature control and crack prevention in mass concrete.

The hydration rate of concrete is influenced by internal factors such as cementitious materials dosage and variety and particle size [6,7,8], admixture type and dosage [9,10,11,12], water-binder ratio [9], aggregate type and dosage [5,13], and admixtures [14], as well as external factors including initial concreting temperature and ambient temperature [15,16,17]. Studies have shown that the initial concreting temperature and the water-binder ratio are two important kinetic factors affecting the hydration rate of concrete [3,13]. In recent years, there have been many studies on the hydration rate of concrete at home and abroad. Freiesleben Hansen et al. proposed the equivalent age based on hydration degree [18]; Zhang Ziming indicated that the effect of temperature on the hydration rate should be considered in the calculation of the equivalent age [19]; and the authors of [7] researched *The Influence of Temperature on the Hydration Rate of Cements Based on Calorimetric Measurements*, indicated that the hydration rate in relation to the curing temperature changes with the progression of hydration. The authors of [20] researched the early hydration process of three types of concrete with a mass ratio of water to binder of 0.4, including ordinary Portland cement concrete (OPC), concrete with fly ash of 30% (FAC), and concrete with silica fume of 5% (SFC). The early hydration process is divided into five stages, which are ion dissolution, induction, acceleration, transition, and deceleration. Incorporation of fly ash reduces the peak of the hydration rate and delays the second hydration acceleration, whereas the addition of 5% silica fume has little effect on the early hydration of concrete. The authors of [21] researched the gas constant (E-a/R), which obviously changes with the hydration degree without a fixed form; the author of [22] researched the mathematical formula that is based on the combination of three rate functions of reaction that represent the effect of moisture condition as well as temperature. These studies on hydration rate mainly focus on the quantification of equivalent age and maturity, the relationship between hydration rate and age, or the relationship between hydration rate and the hydration degree of cementitious materials. However, there are few studies on quantitative models that integrate the effects of various factors on the hydration rate from a dynamic perspective, which in turn affects the accuracy of the concrete adiabatic temperature rise model.

This study was conducted on fly ash concrete of dams to study the effects of initial temperature and water-binder ratio on the hydration process [23] and hydration rate of fly ash concrete through an adiabatic temperature rise test of concrete and calculating kinetic parameters based on the Krstulovic-Dabic model to reveal the microscopic mechanism of hydration reaction, and then the effect mechanism of the initial concreting temperature and water-binder ratio on chemically bound water and pore space of fly ash concrete was researched by a thermogravimetric analyzer and industrial CT scanning techniques.

## 2. Experiment Scheme

### 2.1. Raw Materials

The type 42.5 ordinary silicate cement used in the test was produced by Liquan Conch Co., Ltd.; the Class F Grade II fly ash was produced by Yuanhui Fly Ash Co., Ltd. in Xi’an (China), with an apparent density of 2.5 g/cm^3^; the fine aggregate was river sand with a fineness modulus of 2.61 and an apparent density of 1560 Kg/m^3^; the coarse aggregate was crushed stone with a continuous grading of 5~37.5 mm and an apparent density of 2700 kg/m^3^; the common naphthalene water reducing agent and tap water were used. The properties of cement and fly ash are shown in Table 1 and Table 2, respectively.

### 2.2. Experiment Cases

According to DL/T5330-2015 [24] in Code for Mix Design of Hydraulic Concrete, the proportion ratio was designed by the volume method, the water-binder ratio was set as 0.4, 0.45, 0.5, 0.55, and 0.6, respectively, and the admixture of water-reducing agent was 0.75%; according to SL677-2014 [25] in Specifications for Hydraulic Concrete Construction, the initial concreting temperature of concrete was set at 5 °C, 15 °C, and 25 °C; considering the current fly ash admixture of concrete in hydraulic buildings is between 35% and 45%, the amount of fly ash admixture was set at 35%. The mixed proportion of concrete is detailed in Table 3. The target density of fly ash concrete is 2300–2400 Kg/m^3^.

### 2.3. Experiment Procedures and Their Implementation

Based on the seven groups of proportions designed in Table 3, each component was weighed and mixed into concrete for adiabatic temperature rise, TG-DTG, and industrial CT tests. The initial concreting temperature of the mixture was achieved by placing the raw materials in a thermostat for heating or by adding and mixing cold water.

(1) Adiabatic temperature rise test: in accordance with JG/T329-2011 [26] in Code for Apparatus of concrete thermal coefficient, the test was conducted using HR-4A concrete thermophysical parameter measuring instrument (test temperature was 0 °C~85 °C, temperature resolution was 0.01 °C, and temperature accuracy was 0.01 °C, see Figure 1a), and the mixture was mixed by the machine and then molded into the specimen (the test sample barrel was a cylindrical iron cylinder with a bottom diameter of 400 mm, a height of 400 mm, and a volume of 50.24 L, see Figure 1b). The specimen was first loaded into a square adiabatic box with a side length of 1 m (see Figure 1c) and then quickly fed into the thermophysical parameter tester after the sensor was inserted in its center. The central adiabatic temperature rise was automatically collected by computer at 1 h intervals for 28 days.

(2) TG-DTG test: 100 × 100 × 100 mm cubic specimens were prepared; see Figure 2a. The specimens with initial concreting temperatures of 5 °C and 15 °C were naturally cured according to the measured temperature and humidity curves shown in Figure 3; the specimens with an initial concreting temperature of 25 °C were cured in an environmental maintenance box (model GT-TH-S-225Z; temperature and humidity range: temperature −40 °C~150 °C; humidity 20%~98%RH; control accuracy: temperature ± 0.3 °C; and humidity ± 1.5% RH, see Figure 2b) with a curing temperature of 25 °C and humidity of 75%. The specimens that were cured for 45 min, 12 h, 1 d, 3 d, and 7 d were destroyed to take their central parts. After terminating the hydration with an anhydrous ethanol solution, the samples were powdered (passed through a 0.15 mm square hole sieve) and dried in an oven at 40 °C~50 °C until constant weight. The (45 ± 2) mg sample was placed in a crucible and heated from 20 °C to 1000 °C at a temperature rise rate of 20 °C/min in a nitrogen atmosphere for a thermal analysis test. The test equipment was TGA550 made in the United States (maximum temperature to 1000 °C; temperature accuracy of ± 1 °C; temperature precision of up to 0.01 °C; and temperature rise rate range of 0.1~100 °C/min); see Figure 2c.

(3) Industrial CT test: the specimens were prepared and cured as in the TG-DTG test. The specimens after curing to 1 day, 3 day, and 7 day ages were scanned by industrial CT (model: MultiscaleVoxel-450 industrial CT, sample size ≤ 300 mm, effective detection height ≤500 mm, 3072 × 3072 pixels, 16 bit, pixel size 139 um; effective imaging area 427 mm × 427 mm, see Figure 2d). The CT-scanned images were then reconstructed in three dimensions using VoxelStudio Recon software V2.5.1.25 to provide basic data for analyzing the dynamic changes of pores inside the concrete.

### 2.4. Results and Analysis of the Adiabatic Temperature Rise Test

The adiabatic temperature rise curves of concrete under the influence of the initial concreting temperature and water-binder ratio are shown in Figure 4. All curves in Figure 4 showed rapid growth in the early stage and a gentle rise in the late stage. With the increase in the initial concreting temperature and the decrease in the water-binder ratio, the adiabatic temperature increased, and the adiabatic temperature rise reached its peak at about 700 h (28 d).

The adiabatic temperature rise rate curves obtained by deriving the adiabatic temperature rise data for concrete are shown in Figure 5.

① The hydration temperature rise rate of the specimens increased monotonically to the peak, declined, and gradually leveled off close to a horizontal line after 48 h of age. All specimens went through four periods: an induction period, an acceleration period, a deceleration period, and a decay period. Initially, the rapid hydration of cementitious materials improved the temperature rise rate, followed by a period of rest, which was the induction period of the hydration reaction. The induction periods of the seven groups of specimens ended at 2.8 h, 3.2 h, 3.6 h, 4.0 h, 5.0 h, 6.5 h, and 2.9 h, respectively, with temperature rise rates of 0.751 °C/h, 0.590 °C/h, 0.437 °C/h, 0.425 °C/h, 0.418 °C/h, 0.394 °C/h, and 0.947 °C/h; after that, the specimens entered the acceleration period, and the peak points of the seven groups of specimens appeared at 8 h, 11 h, 12 h, 12.8 h, 16 h, 18 h, and 6 h, respectively, with temperature rise rates of 1.513 °C/h, 1.403 °C/h, 1.301 °C/h, 1.222 °C/h, 1.163 °C/h, 0.796 °C/h, and 1.746 °C/h; then, the hydration rate decreased to the point where the rate changed significantly, which indicated that the specimens entered the decay period. The decay period of the seven groups of specimens occurred at 18 h, 21 h, 24 h, 25 h, 27 h, 30 h, and 18 h, with temperature rise rates of 0.814 °C/h, 0.658 °C/h, 0.588 °C/h, 0.587 °C/h, 0.417 °C/h, 0.418 °C/h, and 0.773 °C/h;

② The end point of each hydration phase was advanced, and the rate of temperature rise was increased by increasing the initial concreting temperature. Compared with that of the specimen with the initial concreting temperature of 5 °C, the end of the induction period, the peak point, and the beginning of the decay period of the specimen with the initial concreting temperature of 25 °C were about 55% (3.6 h), 66% (12 h), and 40% (12 h) earlier, respectively, and the maximum temperature rise rates at these three points were about 140%, 119%, and 84.9% higher, respectively;

③ The end point of each hydration phase was advanced, and the rate of temperature rise was increased by decreasing the water-binder ratio. Compared with that of the specimen with the water-binder ratio of 0.6, the end of the induction period, the peak point, and the beginning of the decay period of the specimen with the water-binder ratio of 0.4 were about 44% (2.2 h), 50% (8 h), and 33% (9 h) earlier, respectively, and the maximum temperature rise rates at these three points were increased by about 79%, 30%, and 95%, respectively.

## 3. Hydration Kinetic Analysis

### 3.1. Hydration Kinetic Analysis Method

#### 3.1.1. Basic Model

According to the Krstulovic–Dabic model [27], the hydration of cementitious materials was divided into three stages: crystal nucleation and growth (NG), phase boundary reaction (I), and diffusion (D). The hydration reaction process depended on the slowest rate among the NG, I, and D. The kinetic model of hydration is expressed as:

NG stage:(1)$${\left[-\ln\left(1-\alpha \right)\right]}^{1/n}=K_{1}\left(t-t_{0}\right)=K_{1}^{\prime }\left(t-t_{0}\right)$$

I stage:(2)$${\left[1-{\left(1-\alpha \right)}^{1/3}\right]}^{1}=K_{2}r^{-1}\left(t-t_{0}\right)=K_{2}^{\prime }\left(t-t_{0}\right)$$

D stage:(3)$${\left[1-{\left(1-\alpha \right)}^{1/3}\right]}^{2}=K_{3}r^{-2}\left(t-t_{0}\right)=K_{3}^{\prime }\left(t-t_{0}\right)$$

Differential form of the NG stage:(4)$$\frac{d\alpha }{dt}=F_{1}\left(\alpha \right)=K_{1}^{\prime }n\left(1-\alpha \right){\left[-\ln\left(1-\alpha \right)\right]}^{1-1/n}$$

Differential form of the I stage:(5)$$\frac{d\alpha }{dt}=F_{2}\left(\alpha \right)=3K_{2}^{\prime }{\left(1-\alpha \right)}^{2/3}$$
where *K_i_*—reaction rate constants (*i* = 1 for NG, *i* = 2 for I, and *i* = 3 for D);

*K_i_*′—apparent rate constant (*i* = 1~3);

*t*_0_—end time of the induction period;

*t*—hydration time;

α—hydration degree.

Differential form of the D stage:(6)$$\frac{d\alpha }{dt}=F_{3}\left(\alpha \right)=3K_{3}^{\prime }{\left(1-\alpha \right)}^{2/3}/\left[2-2{\left(1-\alpha \right)}^{1/3}\right]$$

*n*—reaction order;

*r*—diameter of particles involved in the reaction;

$F_{i}\left(\alpha \right)$—reaction mechanism function (*i* = 1~3).

To express the results of the adiabatic temperature rise test of concrete in terms of the hydration degree or hydration rate of the kinetic model, the hydration kinetic equation proposed by Knudson was adopted [28]:(7)$$\frac{1}{P}=\frac{1}{P_{max}}-\frac{t_{50}}{P_{max}\left(t-t_{0}\right)}$$
where *P*—characteristic value of hydration, taking the value of the adiabatic temperature rise *θ*;

*t*_50_—reaction time required to reach 50% hydration (half-life);

*t* − *t*_0_—hydration time calculated from the end of the induction period.

According to the definition of hydration degree and hydration reaction rate:(8)$$\alpha \left(t\right)=\frac{\theta \left(t\right)}{\theta_{max}}$$
(9)$$\alpha \left(t\right)=\frac{\theta \left(t\right)}{\theta_{max}}\frac{d\alpha }{dt}=\frac{d\theta }{dt}\frac{1}{\theta_{max}}$$
where $\theta \left(t\right)$—adiabatic temperature rise of concrete at time *t*, °C;

$\theta_{max}$—final adiabatic temperature rise, °C;

dθ/dt—rate of adiabatic temperature rise.

#### 3.1.2. Plotting Method of dα/dt~$\alpha \left(t\right)$ Curve

The P in Equation (7) was replaced by the data $\theta \left(t\right)$ from the adiabatic temperature rise test, and the maximum adiabatic temperature rise $\theta_{max}$ and half-life *t*_50_ were obtained by linear fitting. Substituting the maximum adiabatic temperature rise $\theta_{max}$ into Equations (8) and (9), the hydration degree $\alpha \left(t\right)$ and the hydration reaction rate dα/dt at any moment *t* could be calculated.

The kinetic parameters *n*, *K*_1_′, *K*_2_′, and *K*_3_′ could be obtained by substituting the hydration degree $\alpha \left(t\right)$ into Equations (1)–(3), taking logarithms for both sides, and applying the least squares method to fit the ln[−ln(1 − *α*)]~ln(*t* − *t*_0_) curve.

By substituting the kinetic parameters *n*, *K*_1_′, *K*_2_′, and *K*_3_′ into Equations (4)–(6), the relationship between the hydration reaction rate dα/dt and the hydration degree $\alpha \left(t\right)$ for the three stages of crystal nucleation and growth (NG), phase boundary reaction (I), and diffusion (D) were obtained, respectively, and then the relationship curves dα/dt~$\alpha \left(t\right)$ could be drawn.

### 3.2. Hydration Kinetic Analysis of Test Results

#### 3.2.1. Hydration Reaction Rate Curve

According to the results of the adiabatic temperature rise test (Figure 4), and applying Equations (4)–(6), the hydration reaction rate curves of concrete with different initial concreting temperatures (shown in Figure 6a–c) and with different water-binder ratios (shown in Figure 6b,d–g were obtained).

The intersection points *α*_1_ and *α*_2_ of the curves in Figure 6 indicated the hydration degree of the characteristic points of the transition from the crystal nucleation and crystal growth stage (NG) to the phase boundary reaction stage (I) and from the phase boundary reaction stage (I) to the diffusion process (D), respectively.

By analyzing the curves shown in Figure 6, the following laws were derived:

② At the initial concreting temperatures of 5 °C and 15 °C and water-binder ratios of 0.6, 0.55, and 0.5, when $\alpha \left(t\right)<\alpha_{1}$, the dα/dt~$\alpha \left(t\right)$ curve was close to the NG curve; when $\alpha_{1}\leq \alpha \left(t\right)\leq \alpha_{2}$, the dα/dt~$\alpha \left(t\right)$ curve was close to the I curve; and when $\alpha_{2}<\alpha \left(t\right)<1.0$, the dα/dt~$\alpha \left(t\right)$ curve was close to the D curve, indicating that the hydration reaction of all specimens showed three stages of NG-I-D, and the whole reaction process was controlled by the lowest rate;

③ At an initial concreting temperature of 25 °C and water-binder ratios of 0.45 and 0.4, $\left|\alpha_{1}-\alpha_{2}\right|\approx 0$, and the hydration reaction underwent only two stages, NG-D, and without an obvious phase boundary reaction process (I).

#### 3.2.2. Analysis of Kinetic Parameters of the Hydration Reaction

According to the test results and applying the principal methods in Section 3.1, the kinetic parameters of the hydration reaction of concrete were obtained, as detailed in Table 4.

① Effect of initial concreting temperature

With the increase in initial concreting temperature, the hydration reaction order *n* of concrete decreased and the reaction rate constants *K*_1_′, *K*_2_′, and *K*_3_′ increased, indicating that the increase in initial concreting temperature accelerated the hydration reaction rate. At any initial concreting temperature, *K*_1_′ ≈ 3 *K*_2_′ ≈ 9 *K*_3_′, indicating that the hydration reaction rate in the NG process was much higher than that in the I and D processes, which was due to the fact that the NG process was in the acceleration phase, the I process was at the end of the acceleration phase to the deceleration phase, and the D process was in the stabilization phase. When the initial concreting temperature was increased from 5 °C to 15 °C and 25 °C, *n* decreased by 8.37% and 10.76%, *K*_1_′ increased by 8.29% and 6.33%, *K*_2_′ increased by 9.24% and 10.66%, and *K*_3_′ increased by 4.49% and 8.60%, respectively. The accelerated phase boundary reaction was more significantly affected. The reason is that the hydration reaction of cementitious material in the NG process is an autocatalytic reaction, and the I process is affected by temperature. The increase in temperature accelerates the hydration of cementitious material. The research results in the literature [29] also show that the temperature accelerates hydration in the later stage;

② Effect of water-binder ratio

With the decrease in water-binder ratio, the hydration reaction order *n* of concrete decreased, the reaction rate constants *K*_1_′, *K*_2_′, and *K*_3_′ increased, and *K*_1_′ ≈ 2 *K*_2_′ ≈ 9 *K*_3_′, indicating that the reaction rate of the NG process was much higher than that of the I and D processes, which was due to the fact that the NG process was in the acceleration phase, the I process was at the end of the acceleration phase to the deceleration phase, and the D process was in the stabilization phase. When the water-binder ratio was reduced from 0.6 to 0.55, 0.5, 0.45, and 0.4, the reaction order *n* decreased by 5.9%, 11.1%, 14.4%, and 25.8%, the reaction rate constants *K*_1_′ increased by 11.3%, 20.0%, 26.2%, and 29.7%, *K*_2_′ increased by 15.5%, 28.3%, 45.7%, and 67.9%, and *K*_3_′ increased by 11.1%, 29.1%, 54.1%, and 94.4%, respectively. The increase in *K*_2_′ and *K*_3_′ was larger than that in *K*_1_′, of which *K*_3_′ was the most obvious, which indicated that the hydration reaction of the D process was most significantly influenced by the decrease in water-binder ratio. The reason was that the decrease in water-binder ratio led to insufficient water required for hydration of unit cementitious material in the crystallization nucleation and crystal growth process (NG), so the hydration rate was relatively slow, while in the diffusion process (D), the fly ash was excited by calcium hydroxide, a hydration product, to carry out the hydration reaction again, which improved the hydration rate. The study results and the authors of [30] also show that the low water-binder ratio has a great influence on the hydration process.

In short, based on the results of the hydration reaction kinetic parameters *K*_1_′, *K*_2_′, and *K*_3_′, combined with the relationship between the hydration kinetic parameters of cementing materials determined by the authors of [31], it is found that the hydration reaction process of fly ash concrete is different from that of cementing materials, and the early speed is significantly reduced, mainly because the aggregate in concrete will reduce the temperature rise of concrete in the hydration process, which in turn slows down the hydration reaction.

## 4. Thermogravimetric Analysis and CT Scanning Analysis

### 4.1. TG-DTG Analysis

The TG test results of fly ash concrete with different initial concreting temperatures and different water-binder ratios at an age of 1 d are shown in Figure 7 and Figure 8. The TG and DTG curves at a water-binder ratio of 0.45 and an initial concreting temperature of 15 °C are shown in Figure 9. According to the literature [32] and analyzing Figure 7, Figure 8 and Figure 9, it was known that the weight loss from room temperature to 190 °C corresponded to the dehydration decomposition of C-S-H and AFt and the water loss of monocarbon hydrated carboaluminate; the weight loss from 380 °C to 460 °C corresponded to the decomposition of calcium hydroxide; and that from 600 °C to 700 °C corresponded to the decomposition of calcium carbonate.

Considering that the free water in the samples might not be dried sufficiently, the mass loss fraction of the samples was used to calculate the chemically bound water of the samples from 60 °C to 700 °C. The calculation results of the chemically bound water and its change rate are shown in Table 5. The relationship between the chemically bound water and the initial concreting temperature as well as the water-binder ratio are shown in Figure 10 and Figure 11.

According to Figure 10 and Figure 11 and Table 5, the following laws can be derived:

① The content of chemically bound water increased with decreasing initial concreting temperature, with a small increase before 1 d and a large increase after 1 d. The main guess is that the effect of alkali excitation of late hydration products on hydration rate exceeds the effect of temperature on hydration rate, especially resulting in the chemically bound water content at the initial concreting temperature of 5 °C in the late stage exceeding that at 25 °C. Similar conclusions were obtained by Li Jinxing in his study of the effect of temperature and pH of the initial exciter solution on the hydration reaction of slag [33]. The difference between the chemically bound water content at the initial concreting temperatures of 15 °C and 25 °C was much smaller than that at 15 °C and 5 °C. The increase in chemically bound water was significant when the initial concreting temperature was reduced from 15 °C to 5 °C. Equation (10) is the fit of the increase in chemically bound water Δw versus the decrease in initial concreting temperature $\Delta T_{0}$ and age τ with a fitting degree R = 0.952.
(10)$$\Delta w\left(\tau ,\Delta T_{0}\right)=\frac{\tau \times \Delta T_{0}}{C_{1}\times C_{2}{}^{\Delta T_{0}}\times \tau^{C_{3}}}$$
where Δw denotes the increase in chemically bound water (%); τ denotes age (d); and $\Delta T_{0}$ denotes the value-added of initial concreting temperature appreciation (°C); the test coefficients *C*_1_, *C*_2_, and *C*_3_ are fitted by the least squares method as 39.18, −0.96, and 0.75;

② The content of chemically bound water increased with increasing water-binder ratio, mainly because the relatively large water-binder ratio in the cementitious system with mineral admixture made it easier to disperse hydration products, which provided more interface and space needed for hydration, resulting in increased adequate hydration, hydration products, and hydration rate. Similar results were obtained by Yan Peiyu in his study of the effect of water-binder ratio on the hydration of cementitious materials [34]. The increase in chemically bound water was more significant when the water-binder ratio was increased from 0.5 to 0.6 than from 0.4 to 0.5. Equation (11) is the fit of the increase in chemically bound water Δw versus the increase in water-binder ratio Δr and age τ with a fitting degree R = 0.970.
(11)$$\Delta w\left(\tau ,\Delta r\right)=C_{1}+C_{2}\tau +C_{3}\Delta r^{3/2}$$
where Δw denotes the increase in chemically bound water (%); τ denotes age (d); Δr denotes the increase in water-binder ratio; the test coefficients *C*_1_, *C*_2_, and *C*_3_ are fitted by least squares as −0.054, 0.039, and 31.24;

③ The content of chemically bound water increased with age at all initial concreting temperatures and water-binder ratios, with the growth rate being fast from 45 min to 1 d and then gradually slowing down because the fly ash with low reactivity needed to be hydrated under the alkali excitation and it did not participate in hydration in the early stage, resulting in sufficient moisture and rapid hydration in this period. Similar results were obtained by Yan Peiyu [34] in his study on the effect of water-binder ratio on the hydration of cementitious materials. Comparing the differences in the growth rate of chemically bound water in the four time periods of 45 min~12 h, 12 h~1 d, 1 d~3 d, and 3 d~7 d at the water-binder ratio of 0.4 and 0.6, it was concluded that the growth rate of chemically bound water at the water-binder ratio of 0.6 was 1.53 times, 1.9 times, 5.54 times, and 13.8 times than that at the water-binder ratio of 0.4; comparing the differences in the growth rate of chemically bound water in the above four time periods at the water-binder ratio of 0.5 and the initial concreting temperatures of 5 °C and 25 °C, it was concluded that the growth rate of chemically bound water at the initial concreting temperature of 5 °C was 1.3 times, 1.23 times, 1.7 times, and 1 time of that at 25 °C. It was found that the initial concreting temperature of fly ash concrete had a significant effect on the hydration rate from 1 d to 3 d, and the water-binder ratio had a significant effect on the hydration rate from 3 d to 7 d, which was consistent with the results of the hydration kinetic model.

### 4.2. Industrial CT Scanning Analysis

#### 4.2.1. CT Scanned Images

Industrial CT is capable of scanning the specimen layer by layer. In this study, the scanned images of the section at 20 mm depth of the specimens with an age of 1 d were examined for analysis, as shown in Figure 12 and Figure 13, where the black dots represent the pores and their size indicates the size of the pore aperture.

From Figure 12, it can be seen that with the increase in initial concreting temperature, the number of small pores decreased and the large pores increased at 1 d; from Figure 13, it can be seen that with the increase in water-binder ratio, the number of small pores decreased and the large pores increased at 1 d.

#### 4.2.2. Porosity Analysis

Based on CT-scanned images, the geometric features of the pores were reconstructed by VoxelStudio Recon software V2.5.1.25 to quantify them. The results of porosity and its change rate in fly ash concrete at different initial concreting temperatures and different water-binder ratios are shown in Table 6, and the relationship between porosity and age of fly ash concrete at different initial concreting temperatures and different water-binder ratios is shown in Figure 14.

As can be seen from Figure 14, the porosity decreased with decreasing initial concreting temperature, decreasing water-binder ratio, and increasing age. The porosity was affected by the initial concreting temperatures, mainly because the specimens with initial concreting temperatures of 5 °C and 15 °C were cured under variable temperature conditions. Similar results were obtained by Liu Jun in his study on the reduction of porosity by low and variable temperature maintenance conditions, and his study also showed that concrete at low temperatures had low porosity and its strength was not necessarily high [35].

Combined with Figure 14 and Table 6, the following laws were derived:

① The rate of porosity reduction was influenced by the initial concreting temperature and the water-binder ratio. The porosity was reduced by 21~26.4% when the initial concreting temperature was lowered from 15 °C to 5 °C and by 4.9~9.6% from 25 °C to 15 °C; the porosity was reduced by 47.4~51.1% when the water-binder ratio was decreased from 0.45 to 0.4, by 11.1~12.6% from 0.6 to 0.55, and by 5.0~10% from 0.5 to 0.45 as well as from 0.55 to 0.5. It was shown that low and variable temperature curing and a low water-binder ratio were effective in reducing porosity;

② The porosity decreased rapidly during 1 d to 3 d, with the fastest decrease at the initial concreting temperature of 25 °C. When the initial concreting temperature was 25 °C, 15 °C, and 5 °C, the rate of porosity reduction during 1 d to 3 d was 3.25 times, 2 times, and 1.3 times of that during 3 d to 7 d, respectively; when the water-binder ratio was 0.4, the rate of porosity reduction during 1 d to 3 d was 3 times of that during 3 d to 7 d; the rate of porosity reduction during 1 d to 3 d was 1.5~2 times of that during 3 d to 7 d in the case of the other water-binder ratio. It was shown that 1 d to 3 d was the best period to reduce the porosity.

#### 4.2.3. Pore size Analysis

The variation of the proportion of different pore sizes at different initial concreting temperatures and water-binder ratios is shown in Table 7 and Figure 15. In the early stage of hydration, the proportion of pore size r < 0.1 mm was about 10%, the proportion of 0.1 mm < r < 0.5 mm was about 10%~18%, and the proportion of r > 0.5 mm was more than 70%. With the growth of age, the proportion of pore size r < 0.1 mm was increasing, and the smaller the water-binder ratio, the larger the proportion; the proportion of 0.1 mm < r < 0.5 mm was basically unchanged, but the proportion was slightly higher when the initial concreting temperature was relatively low; the proportion of r > 0.5 mm was decreasing.

As shown in Figure 15a–c, the proportion of pore size r < 0.1 mm increased with the decrease in water-binder ratio and the increase in age, among which the growth rate was relatively fast from 1 d to 3 d and when the water-binder ratio was 0.4, and the growth rate tended to slow down from 3 d to 7 d; the pattern of the proportion of r < 0.1 mm in relation to the initial concreting temperature was not obvious; the proportion of porosity at the initial concreting temperature of 25 °C was the smallest, and the growth rate increased with age; the proportion of porosity at the initial concreting temperature of 5 °C was the smallest, but the growth rate with age was slow and basically unchanged.

As shown in Figure 15d, the proportion of pore size 0.1 mm < r < 0.5 mm remained basically the same with age, and slightly decreased at the water-binder ratio of 0.5 and the initial concreting temperature of 25 °C. The proportion of 0.1 mm < r < 0.5 mm was not obviously related to the water-binder ratio and decreased with the increase in the initial concreting temperature.

As shown in Figure 15, the proportion of r > 0.5 mm decreased gradually with age, but there was no obvious pattern between the pore size ratio and the initial concreting temperature as well as the water-binder ratio. The proportion decreased fastest when the water-binder ratio was 0.5 and the initial concreting temperature was 25 °C.

Combined with Figure 12, Figure 13, Figure 14 and Figure 15 and Table 6, it can be seen that the influence of low water-binder ratio, high initial concreting temperature, and early age on porosity was relatively significant, because the increase in initial concreting temperature slowed down the hydration speed in the early stage and reduced the hydration degree and the number of hydration products; with the growth of age, the increasing hydration products filled the pores with relatively large size, thus reducing the total porosity; the reduction of water-binder ratio accelerated the hydration rate and generated more hydration products to fill the pores with relatively large size, thus reducing the total porosity. Additionally, there was a large difference between the change rate of porosity and that of bonded water under the same conditions; for example, when the water-binder ratio was 0.4 and the initial concreting temperature was 15 °C, the rate of change of porosity was 0.25 times that of bound water during 1 d to 3 d, which indicated that the change of porosity was mainly determined by factors like construction and the external environment but was little affected by chemical reactions.

## 5. Conclusions and Discussion

In this study, the effect of the initial concreting temperature and water-binder ratio on the hydration reaction rate of fly ash concrete at an early age was analyzed from macro and micro perspectives based on the adiabatic temperature rise test and applying the basic principle of hydration kinetics, a TG-DTG analyzer, and industrial CT scanning technology. The main conclusions are as follows:

As the initial concreting temperature increased and the water-binder ratio decreased, the adiabatic temperature rise rate increased. The peak of exothermic rate appeared about 12 h earlier at the initial concreting temperature of 25 °C than that at 5 °C, and the maximum exothermic rate increased by 119%; the peak of temperature rise rate appeared about 8 h earlier at the water-binder ratio of 0.4 than that at 0.6, and the maximum temperature rise rate increased by about 30%;

When the peak point of temperature rise appeared before 12 h, the hydration process was transformed from being controlled by three phases of NG-I-D to two phases of NG-D. The initial concreting temperature had a significant effect on the phase boundary reaction (I) process of fly ash concrete, and the water-binder ratio had a significant effect on the diffusion (D) process;

The chemically bound water increased with the decrease in initial concreting temperature and the increase in water-binder ratio at an early age; it also increased with age, with a fast growth rate at the early stage and a slow growth rate at the final stage. The initial concreting temperature had a significant effect on the growth rate of bound water from 1 d to 3 d, and the water-binder ratio had a significant effect on the growth rate of bound water from 3 d to 7 d;

The porosity increased with the decrease in water-binder ratio and the increase in initial concreting temperature at an early age; it also decreased with age, with a rapid reduction during 1 d to 3 d and a slow reduction during 3 d to 7 d, and the decline rate was about (2.0~13.0) × 10^−4^%/h, where 1 d to 3 d was the key period to reduce the porosity;

The pore size was also obviously affected by the initial concreting temperature and water-binder ratio. The proportion of pore size r < 0.1 mm increased with the decrease in water-binder ratio, and the proportion of 0.1 mm < r < 0.5 mm decreased with the increase in initial concreting temperature; the proportion of r > 0.5 mm was not obviously related to the initial concreting temperature and water-binder ratio.

In this study, the effect of the initial concreting temperature and water-binder ratio on the hydration reaction and temperature rise rate at an early age, as well as the variation of chemically bound water, porosity, and pore size, were studied by the specific materials and proportions. However, a comprehensive quantitative relationship has not yet been established due to many uncertainties in the market, such as the variety of materials, the complexity of the actual engineering environment, and the different construction conditions, so further research on the hydrochemical reaction process of concrete is still needed to establish a generalized mathematical model. Additionally, the mechanical properties of concrete change significantly at an early age, which is important for guiding construction, but no relevant research was involved in this study, so the evolution of the mechanical properties of concrete at an early age is also worth further study.

## Figures and Tables

**Figure 1 materials-16-04286-f001:**
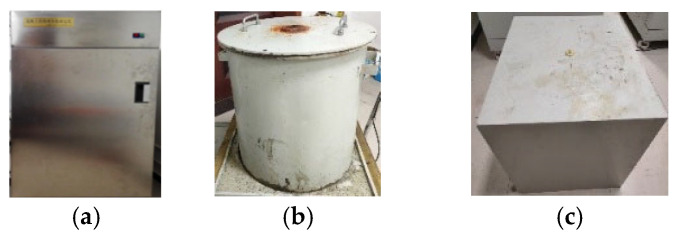
(**a**) Thermophysical parameter tester for concrete; (**b**) test sample barrel; and (**c**) insulation box.

**Figure 2 materials-16-04286-f002:**
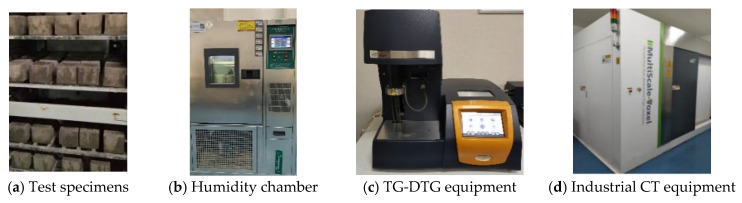
Test specimens and equipment for TG-DTG and industrial CT.

**Figure 3 materials-16-04286-f003:**
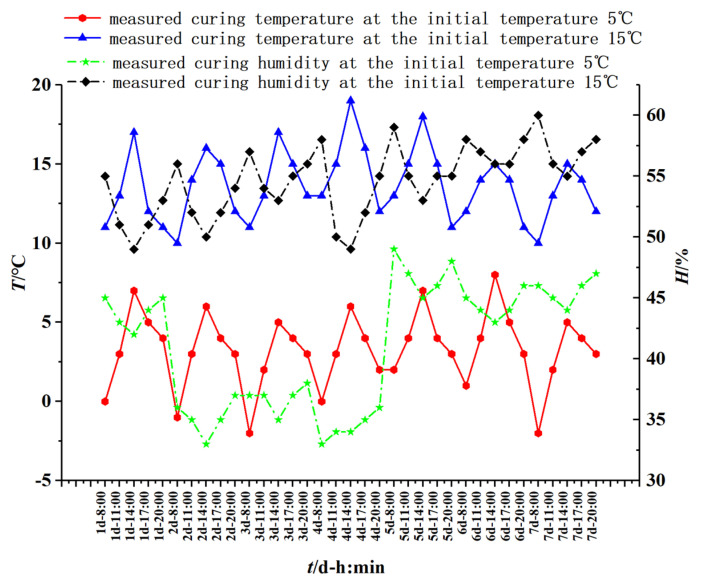
Measured temperature and humidity of natural curing.

**Figure 4 materials-16-04286-f004:**
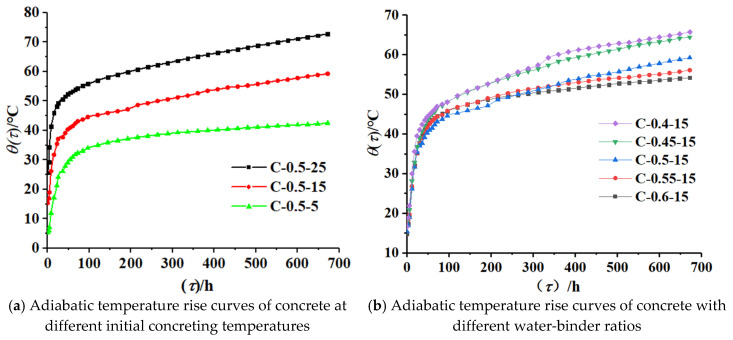
Adiabatic temperature rise curves.

**Figure 5 materials-16-04286-f005:**
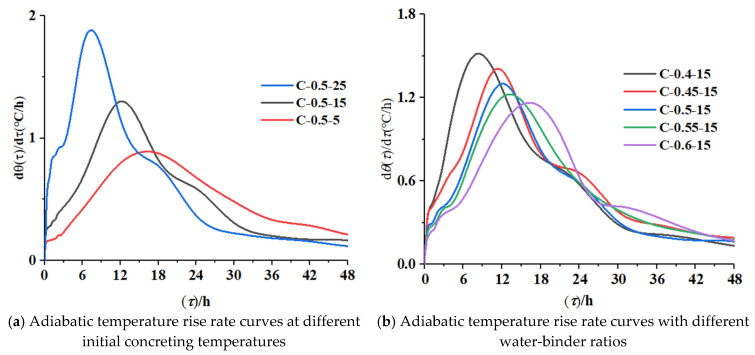
Adiabatic temperature rise rate curves.

**Figure 6 materials-16-04286-f006:**
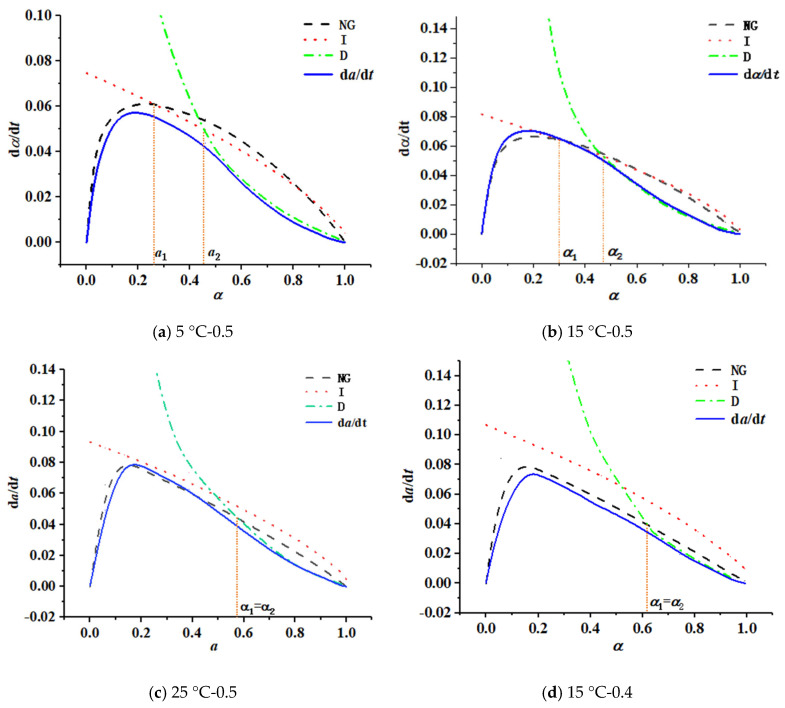
dα/dt~$\alpha \left(t\right)$ curves at different initial concreting temperatures and water-binder ratios.

**Figure 7 materials-16-04286-f007:**
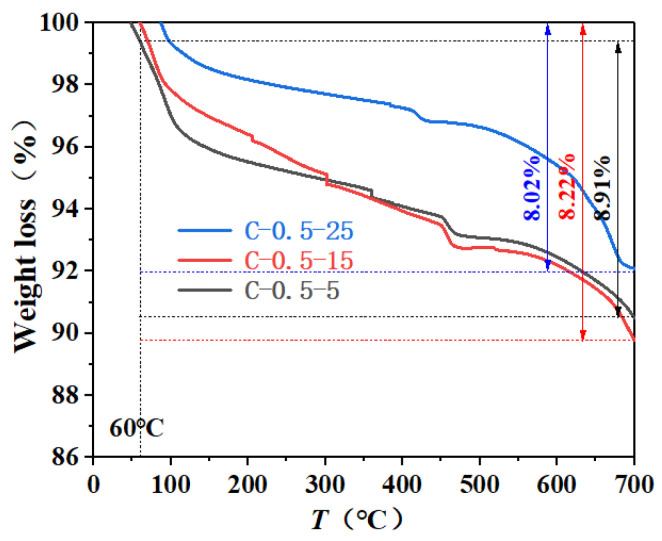
Thermogravimetric TG curves at different initial concreting temperatures (1st d).

**Figure 8 materials-16-04286-f008:**
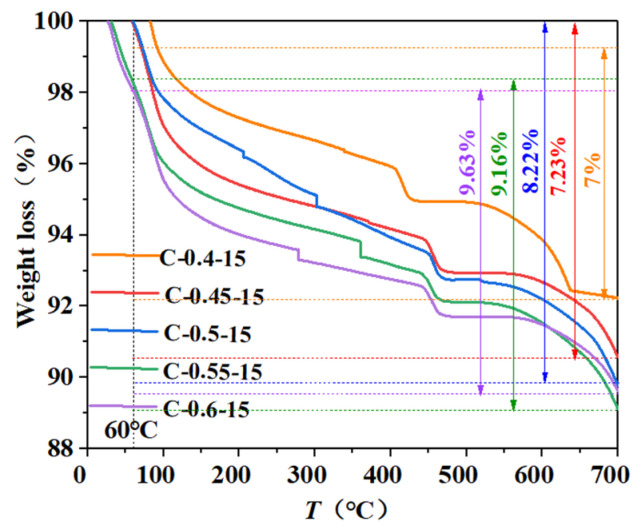
Thermogravimetric TG curves at different water-binder ratios (1st d).

**Figure 9 materials-16-04286-f009:**
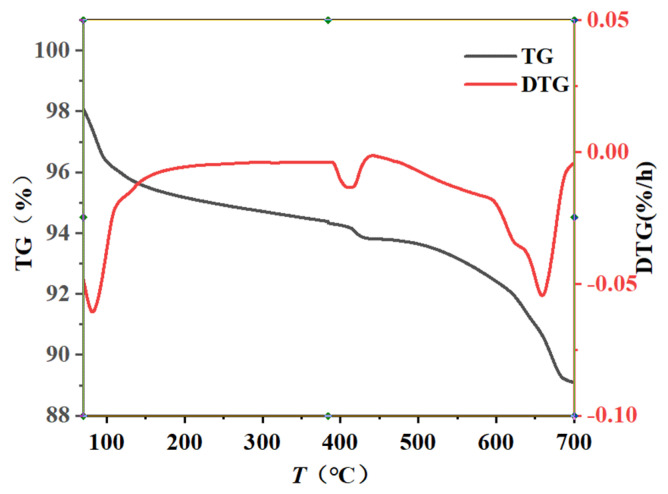
TG and DTG curves at 0.45–15 °C (1st d).

**Figure 10 materials-16-04286-f010:**
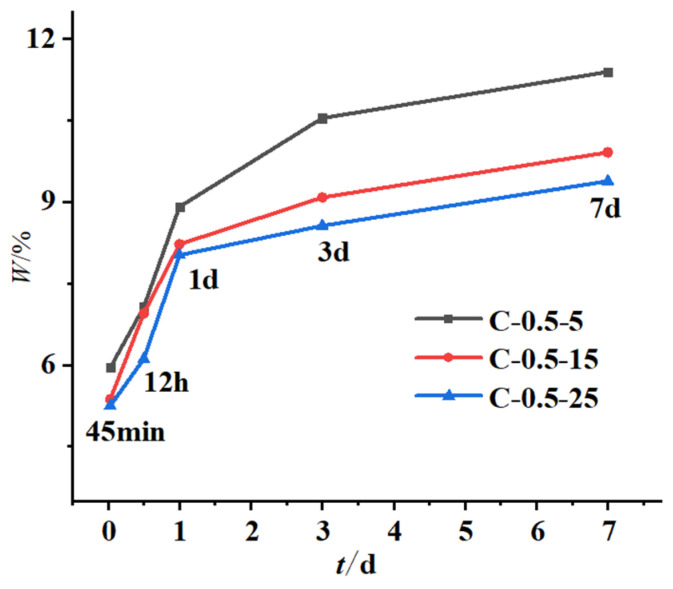
Relationship curves between chemically bound water and initial concreting temperature.

**Figure 11 materials-16-04286-f011:**
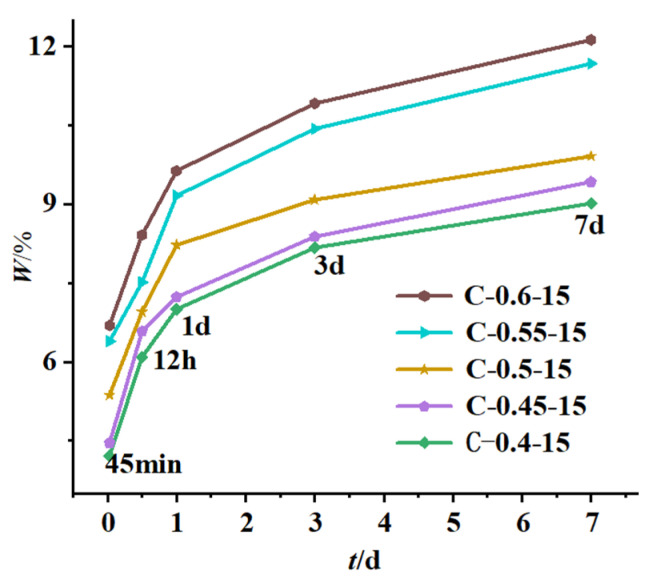
Relationship curves between chemically bound water and the water-binder ratio.

**Figure 12 materials-16-04286-f012:**
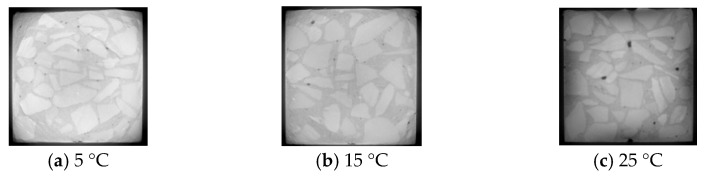
Two-dimensional cross-sectional view of CT with different initial concreting temperatures (1 d, depth 20 mm).

**Figure 13 materials-16-04286-f013:**
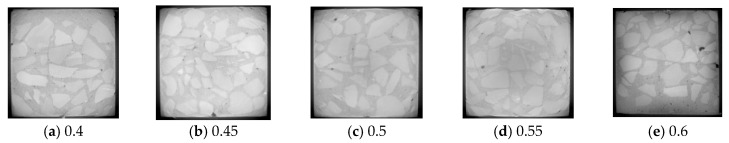
Two-dimensional cross-sectional view of CT with different water-cement ratios (1 d, depth 20 mm).

**Figure 14 materials-16-04286-f014:**
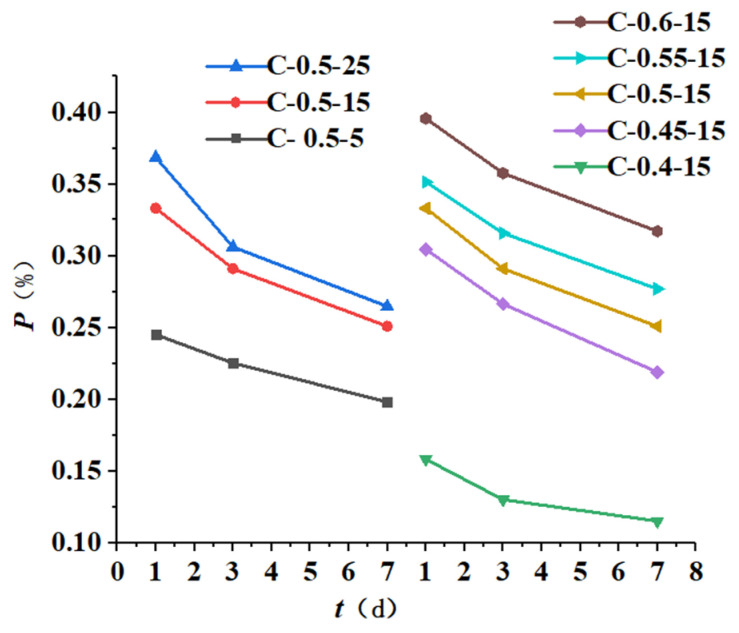
Relationship between porosity and age.

**Figure 15 materials-16-04286-f015:**
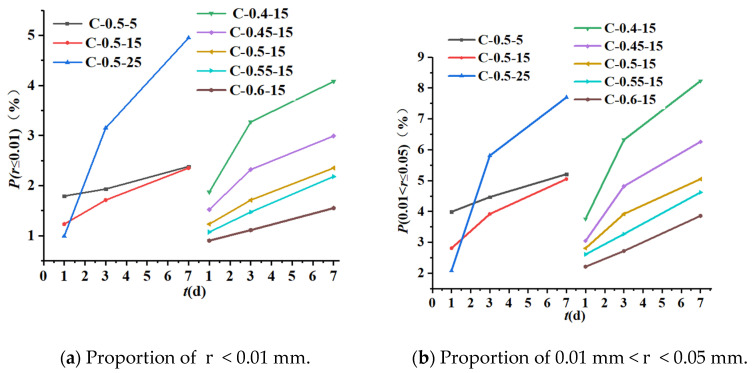
The variation curves of the proportion of different pore sizes r at different initial concreting temperatures and water-binder ratios.

**Table 1 materials-16-04286-t001:** Basic properties of cement.

Density (g/cm^3^)	Specific Surface Area (m^2^/kg)	Compressive Strength (MPa)		Setting Time (min)		Stability	Water Consumption for Standard Consistency (%)
		3 d	28 d	Initial Setting	Final Setting		
3.04	345	29.8	43.2	54	351	Qualified	26.9

**Table 2 materials-16-04286-t002:** Basic properties and chemical composition of fly ash.

Fineness (%)	Water Demand Ratio (%)	Loss on Ignition (%)	Moisture Content (%)	Activity Index (%)	SO_3_ Content (%)	F-CaO Content (%)	Stability
23.1	99	3.7	0.42	74	0.95	0.06	Qualified

**Table 3 materials-16-04286-t003:** The mix proportion of concrete.

Specimen No.	Water-Binder Ratio	Initial Concreting Temperature (°C)	Cement (Kg/m^3^)	Fly Ash (Kg/m^3^)	Water (Kg/m^3^)	Sand (Kg/m^3^)	Stone (Kg/m^3^)	Water Reducing Agent (Kg/m^3^)
C-0.4-15	0.4	15	292.5	236.25	180	473.42	1141.95	8.1
C-0.45-15	0.45	15	260	210	180	505.96	1171.59	8.46
C-0.5-15	0.5	15	234	189	180	530.1	1196.01	7.61
C-0.55-15	0.55	15	212.7	171.79	180	552.61	1215.18	6.92
C-0.6-15	0.6	15	195	157.5	180	568.52	1231.92	6.35
C-0.5-5	0.5	5	234	189	180	530.1	1196.01	7.61
C-0.5-25	0.5	25	234	189	180	530.1	1196.01	7.61

**Table 4 materials-16-04286-t004:** Kinetic parameters of the hydration reaction.

Specimen No.	Test Conditions		*n*	*K*_1_′	*K*_2_′	*K*_3_′	Hydration Process	*α* _1_	*α* _2_
	Initial Concreting Temperature/°C	Water-Binder Ratio							
C-0.4-15	15	0.4	1.0564	0.0982	0.0356	0.0145	NG-D	/	0.62
C-0.45-15	15	0.45	1.2190	0.0955	0.0309	0.0111	NG-D	/	0.48
C-0.5-15	15	0.5	1.2671	0.0909	0.0272	0.0096	NG-I-D	0.30	0.47
C-0.55-15	15	0.55	1.3397	0.0843	0.0245	0.0080	NG-I-D	0.25	0.43
C-0.6-15	15	0.6	1.4245	0.0757	0.0212	0.0072	NG-I-D	0.22	0.41
C-0.5-5	5	0.5	1.3829	0.0835	0.0249	0.0089	NG-I-D	0.25	0.45
C-0.5-25	25	0.5	1.1308	0.0958	0.0311	0.0102	NG-D	/	0.55

Analyzing the kinetic parameters of the hydration reaction in Table 4, the following laws were derived:

**Table 5 materials-16-04286-t005:** Chemically bound water content and its rate of change.

Specimen No.	Initial Concreting Temperature/°C	Water-Binder Ratio	Chemically Bound Water	Age									
				45 min		12 h		1 d		3 d		7 d	
C-0.4-15	15	0.4	$w\left(\%\right)$	4.21		6.09		7		8.17		9.01	
			$dw/dt\left(\%/h\right)$	0.167			0.076		0.024		0.008		
C-0.45-15	15	0.45	$w\left(\%\right)$	4.46		6.58		7.23		8.38		9.42	
			$dw/dt\left(\%/h\right)$	0.188			0.054		0.024		0.011		
C-0.5-15	15	0.5	$w\left(\%\right)$	5.37		6.95		8.22		9.08		9.91	
			$dw/dt\left(\%/h\right)$	0.140			0.106		0.018		0.009		
C-0.55-15	15	0.55	$w\left(\%\right)$	6.39		7.51		9.16		10.43		11.67	
			$dw/dt\left(\%/h\right)$	0.100			0.138		0.026		0.013		
C-0.6-15	15	0.6	$w\left(\%\right)$	6.69		8.41		9.63		10.91		12.12	
			$dw/dt\left(\%/h\right)$	0.257			0.145		0.133		0.111		
C-0.5-5	5	0.5	$w\left(\%\right)$	5.95		7.066		8.91		10.54		11.39	
			$dw/dt\left(\%/h\right)$	0.099			0.154		0.034		0.009		
C-0.5-25	25	0.5	$w\left(\%\right)$	5.25		6.11		8.02		8.56		9.38	
			$dw/dt\left(\%/h\right)$	0.076			0.159		0.011		0.009		

**Table 6 materials-16-04286-t006:** Porosity and pore change rate at different initial concreting temperatures and different water-binder ratios.

Specimen No.	Initial Concreting Temperature	Water-Binder Ratio	Porosity	Age					
				1 d		3 d		7 d	
C-0.4-15	15	0.4	p%	0.1582		0.1301		0.1151	
			dp/dt (%/h)	0.0006			0.0002		
C-0.45-15	15	0.45	p%	0.3045		0.2665		0.2189	
			dp/dt (%/h)	0.0008			0.0005		
C-0.5-15	15	0.5	p%	0.3331		0.2910		0.2508	
			dp/dt (%/h)	0.0008			0.0004		
C-0.55-15	15	0.55	p%	0.3514		0.3158		0.2769	
			dp/dt (%/h)	0.0006			0.0004		
C-0.6-15	15	0.6	p%	0.3957		0.3576		0.3171	
			dp/dt (%/h)	0.0008			0.0004		
C-0.5-5	5	0.5	p%	0.2450		0.2253		0.1981	
			dp/dt (%/h)	0.0004			0.0003		
C-0.5-25	25	0.5	p%	0.3685		0.3062		0.2647	
			dp/dt (%/h)	0.0013			0.0004		

**Table 7 materials-16-04286-t007:** Pore volume ratio at different pore sizes.

Specimen No.			Age			Porosity (%)
		<0.01 mm^3^	0.01 mm^3^–0.05 mm^3^	0.05 mm^3^–0.1 mm^3^	0.1 mm^3^–0.5 mm^3^	>0.5 mm^3^
C-0.4-15	1 d	1.87	3.76	4.06	13.98	76.33
	3 d	3.27	6.32	7.09	12.96	70.36
	7 d	4.08	8.23	9.54	12.18	65.97
C-0.45-15	1 d	1.48	3.01	3.06	10.44	82.01
	3 d	2.02	4.32	4.47	9.63	79.56
	7 d	2.49	5.76	5.89	9.54	76.32
C-0.5-15	1 d	1.23	2.81	2.88	13.43	79.65
	3 d	1.71	3.92	3.97	13.23	77.17
	7 d	2.35	5.05	5.45	13.18	73.97
C-0.55-15	1 d	1.07	2.61	2.76	13.21	80.35
	3 d	1.47	3.27	3.64	13.02	78.60
	7 d	2.18	4.62	4.89	12.68	75.63
C-0.6-15	1 d	0.90	2.21	2.31	11.84	82.74
	3 d	1.11	2.72	2.86	11.79	81.52
	7 d	1.55	3.86	4.05	11.58	78.96
C-0.5-5	1 d	1.79	3.99	4.71	18.30	71.21
	3 d	1.93	4.47	5.24	18.24	70.12
	7 d	2.38	5.21	6.33	18.13	67.95
C-0.5-25	1 d	0.99	2.08	2.46	10.53	83.94
	3 d	3.15	5.81	4.97	9.36	76.71
	7 d	4.95	7.70	7.71	8.64	71.00

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
