# Peer review of "Influence Mechanism of Initial Concreting Temperature and Water-Binder Ratio on Hydration Rate of Fly Ash Concrete"

_materials, 2023, doi:10.3390/ma16124286_

Round 1

Reviewer 1 Report

Title: Influence Mechanism of Initial Concreting Temperature and Water-Binder Ratio on Hydration Rate of Fly Ash Concrete

1-     Paper is already available online at SSRN 4140198 (https://papers.ssrn.com/sol3/papers.cfm?abstract_id=4140198)

2-     Overall, the manuscript presents a thorough analysis of the effect of initial concreting temperature and water-binder ratio on the hydration reaction rate of fly ash concrete at early age. The study employs a range of analytical techniques, including adiabatic temperature rise testing, TG-DTG analysis, and industrial CT scanning, to provide both macro and micro perspectives on the hydration process. The results show that initial concreting temperature and water-binder ratio significantly influence the adiabatic temperature rise rate, chemical bound water, porosity, and pore size of the concrete at early age.

However, there are a few areas where the manuscript could be improved.

3-      Firstly, while the study provides detailed findings on the hydration process, the authors do not establish a comprehensive quantitative relationship between the variables studied.

4-     The authors briefly mention that there are uncertainties in the market that make it difficult to establish a generalized mathematical model for the hydration process. However, they could provide more detail on the specific limitations of their study, such as the range of initial concreting temperatures and water-binder ratios tested, and how these limitations may affect the generalizability of the results.

5-     Secondly, the manuscript does not include any research on the mechanical properties of concrete at early age, which is an important consideration for guiding construction practices.

Overall, the manuscript provides valuable insights into the hydration process of fly ash concrete, and the findings have important implications for optimizing concrete mix designs.

Author Response

Title: Influence Mechanism of Initial Concreting Temperature and Water-Binder Ratio on Hydration Rate of Fly Ash Concrete

1-     Paper is already available online at SSRN 4140198 (https://papers.ssrn.com/sol3/papers.cfm?abstract_id=4140198)

2-     Overall, the manuscript presents a thorough analysis of the effect of initial concreting temperature and water-binder ratio on the hydration reaction rate of fly ash concrete at early age. The study employs a range of analytical techniques, including adiabatic temperature rise testing, TG-DTG analysis, and industrial CT scanning, to provide both macro and micro perspectives on the hydration process. The results show that initial concreting temperature and water-binder ratio significantly influence the adiabatic temperature rise rate, chemical bound water, porosity, and pore size of the concrete at early age.

However, there are a few areas where the manuscript could be improved.

3-      Firstly, while the study provides detailed findings on the hydration process, the authors do not establish a comprehensive quantitative relationship between the variables studied.

Answer:The paper mainly establishes the relationship between the hydration rate of fly ash concrete and the single factor of water-binder ratio and initial temperature. The relationship will be established from the multi-factor aspect later.

4-     The authors briefly mention that there are uncertainties in the market that make it difficult to establish a generalized mathematical model for the hydration process. However, they could provide more detail on the specific limitations of their study, such as the range of initial concreting temperatures and water-binder ratios tested, and how these limitations may affect the generalizability of the results.

Answer:The paper mainly studied the influence law of initial pouring temperature and water-binder ratio on the hydration rate of fly ash concrete, and studied the influence mechanism of its effect on the hydration rate of fly ash concrete through the changes of hydration kinetic parameters, chemically bound water and pore structure.

5-     Secondly, the manuscript does not include any research on the mechanical properties of concrete at early age, which is an important consideration for guiding construction practices.

Answer:The paper mainly studies the hydration rate of fly ash concrete from the aspect of thermal properties, and then studies from the aspect of mechanical properties.

Overall, the manuscript provides valuable insights into the hydration process of fly ash concrete, and the findings have important implications for optimizing concrete mix designs.

Reviewer 2 Report

1-      In Line 79 This study was conducted on fly ash concrete of dams is this study target is for heavy weight concrete as dams…if right why did not authors mention this as obvious or authors aim to find key solution for temperature causing shrinkage….please explain

2-      In line 103 temperature of concrete was set at 5°C, 15°C, and 25°C; how could authors achieve this conditions?

3-      It is better for table 3 to contain target density

4-      What is the reason for choosing OPC not low heat cement?

5-      What is  the concept for design concrete mix with low cement content of 292.5 as maximum content and decreases with increasing sand content

6-      Line 113 Adiabatic temperature rise test: in accordance with JG/T329-2011 ..its required to add citation for all specifications in the manuscript

7-       Figure 3. Measured temperature and humidity of natural curing….show natural curing condition mentioned here

8-      Axis data for all figures are not obvious …increase figures quality

9-      From line 151 to 180 no references mentioned. It seems to be report not scientific research please insert citation to improve paragraph

10-   All discussion needs to be cited to deep discussion as the present is very shallow

1-   All figures needs to be improved

1-   Pore size analysis should be represented in good method or better figures and discussion

Author Response

1 In Line 79 This study was conducted on fly ash concrete of dams is this study target is for heavy weight concrete as dams…if right why did not authors mention this as obvious or authors aim to find key solution for temperature causing shrinkage….please explain

Answer:The purpose of this study is to provide reference for temperature control and crack prevention measures (initial pouring temperature control and water-binder ratio selection) of mass concrete by studying the hydration temperature rise rate and influence mechanism of fly ash concrete.

2-      In line 103 temperature of concrete was set at 5°C, 15°C, and 25°C; how could authors achieve this conditions?

Answer:The initial temperature is selected by choosing the appropriate climate for pouring, and different batches of test are set at different test times. The initial pouring temperature of 5℃ was tested in January, the initial pouring temperature of 15℃ was tested in April, and the initial pouring temperature of 25℃ was tested in August.

3-      It is better for table 3 to contain target density

Answer:The target density is 2300~2400Kg/m3.

4-      What is the reason for choosing OPC not low heat cement?

Answer:Because OPC ordinary Portland cement is one of the most commonly used cement. The paper Want to study the common cement in concrete hydration temperature rise in the process of influence law.

5-      What is  the concept for design concrete mix with low cement content of 292.5 as maximum content and decreases with increasing sand content

Answer:According to the fly ash concrete mix design method (reference [21]), the cement dosage under each formula is calculated according to the cement dosage determined by the benchmark mix ratio multiplied by the proportion of cement to the cementing material.

6 Line 113 Adiabatic temperature rise test: in accordance with JG/T329-2011 ..its required to add citation for all specifications in the manuscript

Answer:The specification[25][26][27] has been cited in this article.

7 Figure 3. Measured temperature and humidity of natural curing….show natural curing condition mentioned here

Answer:FIG. 3 shows the monitoring results of natural curing temperature and humidity of DSC test specimens.

8-      Axis data for all figures are not obvious …increase figures quality

Answer:All the graphs have been modified, and the axis data and axis bars have been modified.

9-      From line 151 to 180 no references mentioned. It seems to be report not scientific research please insert citation to improve paragraph

Answer:From line 151 to 180, This paper mainly analyzes the law of the test results. The following contents will be analyzed, discussed and compared from the aspects of chemically bound water, porosity and pore structure.

10-   All discussion needs to be cited to deep discussion as the present is very shallow

Answer:The mechanism of hydration kinetics parameters and porosity analysis is discussed, and the results are compared with those already studied.

  • All figures needs to be improved

Answer:The figures have all been changed to Times New Roman10.

1-   Pore size analysis should be represented in good method or better figures and discussion

Answer:The pore size analysis is tabulated and 

Round 2

Reviewer 1 Report

Authors have sufficiently revised the manuscripts.

Reviewer 2 Report

the research could be accepted in present form